# Risk screening methods for extreme heat: Implications for equity-oriented adaptation

Lynée L. Turek-Hankins[1,2¤a]*, Miyuki Hino[3,4¤b], Katharine J. Mach[5,6¤a]

1 Stanford Woods Institute for the Environment, Mentoring Undergraduates in Interdisciplinary Research (MUIR) Program, Stanford University, Stanford, CA, United States of America, 2 Environmental Science and Policy Graduate Program, Leonard and Jayne Abess Center for Ecosystem Science and Policy, University of Miami, Coral Gables, FL, United States of America, 3 Department of City and Regional Planning, The University of North Carolina at Chapel Hill, Chapel Hill, NC, United States of America, 4 Environment, Ecology, and Energy Program, The University of North Carolina at Chapel Hill, Chapel Hill, NC, United States of America, 5 Rosenstiel School of Marine and Atmospheric Science, University of Miami, Miami, FL, United States of America, 6 Leonard and Jayne Abess Center for Ecosystem Science and Policy, University of Miami, Coral Gables, FL, United States of America

¤a Current address: Rosenstiel School of Marine and Atmospheric Science, University of Miami, Miami, FL, United States of America
¤b Current address: Department of City and Regional Planning, The University of North Carolina at Chapel Hill, Chapel Hill, NC, United States of America
* lturek@miami.edu

**Data Availability Statement:** All data come from publicly available sources and can be downloaded off of their original websites. Social Vulnerability Index data are from https://svi.cdc.gov/data-and-tools-download.html CalEnviroScreen 3.0 data are

## Abstract

Morbidity and mortality impacts of extreme heat amplified by climate change will be unequally distributed among communities given pre-existing differences in socioeconomic, health, and environmental conditions. Many governments are interested in adaptation policies that target those especially vulnerable to the risks, but there are important questions about how to effectively identify and support communities most in need of heat adaptations. Here, we use an equity-oriented adaptation program from the state of California as a case study to evaluate the implications of the currently used environmental justice index (CalEnviroScreen 3.0) for the identification of socially vulnerable communities with climate change adaptation needs. As CalEnviroScreen is geared towards air and water pollution, we assess how community heat risks and adaptation needs would be evaluated differently under two more adaptation-relevant vulnerability indices: the Social Vulnerability Index and the Heat-Health Action Index. Our analysis considers communities at the census tract scale, as well as the patterns emerging at the regional scale. Using the current index, the state designates 25% of its census tracts as "disadvantaged" communities eligible for special adaptation funds. However, an additional 12.6% of the state's communities could be considered vulnerable if the two other indices were considered instead. Only 13.4% of communities are vulnerable across all three vulnerability indices studied. Choice of vulnerability index shapes statewide trends in extreme heat risk and is linked to a community's likelihood of receiving heat-related California Climate Investments (CCI) projects. Tracts that are vulnerable under the current pollution-focused index, but not under the heat-health specific index, received four times the number of heat-related interventions as tracts vulnerable under the reverse scenario. This study demonstrates important nuances relevant to implementing equity-

from https://oehha.ca.gov/calenviroscreen/report/
calenviroscreen-30 Heat Health Action Index data
are from https://www.cal-heat.org/download
Temperature data are from https://cal-adapt.org/
data/ Humidity data are from https://cal-adapt.org/
data/ California Climate Investments project data
are from https://webmaps.arb.ca.gov/ccimap/ and
https://ww3.arb.ca.gov/cc/capandtrade/
auctionproceeds/communityinvestments.htm.

**Funding:** This research was funded by the
Mentoring Undergraduates in Interdisciplinary
Research (MUIR) Program at the Stanford Woods
Institute for the Environment, the Rosenstiel
School of Marine and Atmospheric Science at the
University of Miami, and the Abess Center for
Ecosystem Science and Policy at the University of
Miami.

**Competing interests:** The authors have declared
that no competing interests exist.

**Abbreviations:** GGRF, Greenhouse Gas Reduction
Fund; CEJA, California Environmental Justice
Alliance; IPCC, Intergovernmental Panel on Climate
Change; CES, CalEnviroScreen 3.0; HHAI, Heat-
Health Action Index; SVI, Social Vulnerability Index;
CCI, California Climate Investments.

oriented adaptation and explores the challenges, trade-offs, and opportunities in quantifying vulnerability.

## Introduction

Extreme heat threatens the health and wellbeing of people across the globe, but heat-related health impacts are not equally shared among communities. Environmental, health, and socio-economic characteristics of communities are linked to differential, and in some cases disproportionate, heat-related morbidity and mortality impacts [1–4]. Some factors magnify a community's exposure to heat hazards, and other characteristics influence the social, physiological, and material capability of residents to respond to and cope with extreme heat [5, 6]. As climate change continues to increase the frequency and severity of extreme heat events, addressing the social and environmental inequities that increase risk in certain neighborhoods will become more urgent.

Many governments, non-governmental actors, and communities are interested in addressing disparities in climate impacts and supporting adaptation for those most at risk. In some places, there are emerging efforts to implement equity-oriented adaptation policies. The state of California is one such example. Currently, the state is mandated by law to delegate 25% of its funds from the Greenhouse Gas Reduction Fund (GGRF) to "disadvantaged" communities identified through a vulnerability index called CalEnviroScreen 3.0 (CES) [7–9]. Because equity-oriented climate adaptation is an emerging priority, this is an especially important moment to evaluate the state's approach and consider alternatives. We explore the implications of using vulnerability indices to guide equity-oriented adaptation by considering three key issues: 1) how assessed vulnerability differs depending on the vulnerability index used, 2) how extreme heat risk compares among communities across definitions of exposure and vulnerability, and 3) how adaptation projects are distributed across communities at risk from extreme heat.

## Background

This study examines the implications of using different quantitative measures of vulnerability for allocating adaptation investments. In particular, we evaluate the relationship among different measures of vulnerability, extreme heat, and adaptation investment. We employ the definitions of vulnerability and exposure used by the Intergovernmental Panel on Climate Change (IPCC). Exposure is "the presence of people, livelihoods, species or ecosystems, environmental functions, services, and resources, infrastructure, or economic, social, or cultural assets in places and settings that could be adversely affected" [10]. Vulnerability is "the propensity or predisposition to be adversely affected. Vulnerability encompasses a variety of concepts and elements including sensitivity or susceptibility to harm and lack of capacity to cope and adapt" [10]. We follow the precedent set by the state of California and the California Environmental Justice Alliance (CEJA) regarding language used to describe environmental justice communities, while recognizing that the terms "vulnerable" and "disadvantaged" can be seen as disempowering and are often not terms preferred by the communities [9, 11–13]. In this study, a "disadvantaged community" is therefore a census tract whose vulnerability index percentile value is in the upper quartile.

### The health impacts of extreme heat

Extreme heat has had devastating impacts globally, and its risks will increase with future warming [10]. Between 1999 and 2018, 8,909 died from extreme heat-related health causes in

the United States [14, 15]. Past heat-health impacts are numerous for the state of California as well. Between 1999 and 2009, there were 15 heat waves in California that together caused over 11,000 hospitalizations [16]. The 2006 heat wave alone sent 16,166 people to the emergency department, hospitalized 1,182 to 1,254 people, and took an estimated 160 to 505 lives [3, 16, 17]. Reported heat-health morbidity and mortality figures are widely believed to be underestimates, particularly among workers [17, 18]. In the United States, it is projected that future heat wave events exacerbated by climate change will lead to anywhere from thousands to tens of thousands of excess-mortality deaths by the end of the century [19–22].

## Differences in extreme heat definitions

There is no uniform definition of an extreme heat event. Diverse local climates make a single heat definition or threshold unfavorable because people who live in hotter climates have different health responses to equivalent levels of heat exposure as their counterparts in cooler climates [3, 4, 23]. Currently the National Weather Service alerts the public of a heat hazard when the heat index is greater than 105℉; for a minimum of two consecutive days [24]. The heat index is a measurement of the apparent temperature one would feel in the shade given the amplifying effects of humidity; it is an absolute metric that does not consider relative climatology and does not capture the fact that health impacts from different doses of heat vary among communities [2, 4, 16, 25]. The heat index is not to be confused with a vulnerability index, which is a metric that integrates indicators of vulnerability. To address the shortcomings of the heat index, many studies also measure extreme heat using a heat wave definition that spans multiple days and uses a relative, locally defined temperature threshold. However, many heat index and heat wave definitions employ extreme heat thresholds above temperatures where heat-related health impacts begin, potentially missing harmful events [16, 25, 26]. This study therefore includes metrics that measure absolute and relative extreme heat at thresholds shown to be relevant to heat-health risk.

## Differences in vulnerability indices

Vulnerability indices are often used to identify which populations are most susceptible to impacts from environmental hazards, but there are important differences and uncertainties across metrics [27]. Previous research demonstrated that data scaling, normalization, transformation, weighting, and aggregation techniques vary widely in index construction and that different design decisions and spatial resolutions impact measured outcomes [28–30]. A commonly applied index-construction technique is principal component analysis and factor weighting. Heat-specific vulnerability indices similarly differ in indicator weightings and selection [26, 31, 32]. The implications of different vulnerability index definitions and measurement methods as well as their consequences in guiding policy and funding are incompletely understood. Multiple studies agree on the importance of supplementing indices and high-level assessments of vulnerability and environmental justice with "ground-truthing," which validates and embeds context into the process [33–35].

California is a vast and diverse state, and it is unlikely that a single vulnerability definition could encompass all nuances relevant across communities and the hazards they face. California's Fourth Climate Change Assessment Climate Justice Report touched on the limitations of CES in its analysis, comparing it to the Healthy Places Index [36]. CEJA released a report focused on the use of CES in various environmental justice projects throughout the state, highlighting recommended uses of CES in policy making and implementation decisions [11]. Finally, an EPA-sponsored Regionally Applied Research Effort project used multiple environmental justice screening indices and ground-truthing methods to develop a regional

understanding of cumulative pollution exposure in California's San Joaquin Valley [33]. This study fortifies and expands on these projects. Furthermore, we aim to assist in the development of "more robust mapping tools," an identified knowledge gap, by strengthening knowledge of potential impacts and biases that could result from using vulnerability indices in policy development and climate adaptation decision making [36].

## Methods

Our analysis of vulnerability, exposure, and adaptation needs for extreme heat comprised three stages. First, we compared the communities identified as "disadvantaged" based on three different vulnerability indices: CalEnviroScreen 3.0 (CES), the Heat-Health Action Index (HHAI), and the Social Vulnerability Index (SVI). Second, we plotted different definitions of extreme heat against these indices to identify statewide relationships between vulnerability and exposure. Third, we assessed how the state has distributed heat-related CCI-funded projects in relation to community vulnerability and identified the efficacy of Assembly Bill 1550 (AB 1550) in increasing investments beyond the CES-designated communities. AB 1550 was passed by the state of California in 2016, following from concerns that low income communities that may not have pollution burden issues were also in need of climate investments [11]. It upholds the requirement that 25% of the GGRF goes to "disadvantaged communities" as defined by the California Environmental Protection Agency and mandates that an additional 10% of the proceeds go to low income households or communities.

### Mapping social vulnerability

We analyzed the level of agreement among three different social vulnerability indices: SVI, HHAI, and CES. SVI is able to "provide state, local, and tribal disaster management personnel information to target for intervention those [census] tracts that may be socially vulnerable before, during, and after a hazard event" [37]. It was included in this analysis because it is the index used by the Center for Disease Control and Prevention (CDC) in determining those most vulnerable to disasters of all forms. HHAI was created in California's Fourth Climate Change Assessment to "identify neighborhoods or areas that are likely to be more susceptible to future [heat-health events]" [26]. It was studied because it is designed to measure heat-specific vulnerability in California. HHAI scores were converted to percentile ranks for ease of comparison to the other indices. Finally, CES aims to identify "communities most burdened by pollution from multiple sources and most vulnerable to its effects, taking into account the socioeconomic and health status of people living in those communities" [7]. Unlike HHAI and SVI, CES was not directly intended to be used in the context of climate risks, although pollution exposure can exacerbate climate change impacts. We analyzed CES because it is the metric that the state is directed to use for designating a community's status as "disadvantaged." To understand the degree of overlap among the different indices, we classified each variable in each index as pertaining to economic, demographic, health, or environmental conditions.

Data on vulnerability indices were obtained from their respective websites [38–40]. We removed tracts that contained NA values for any of the indices; tracts received NA values from the score developers if they had insufficient or unreliable data for multiple variables or if their population was zero. For all indices, the highest percentile values indicated the most vulnerable census tracts. CES considers a tract vulnerable or "disadvantaged" if its score is the 75th percentile or higher [41]. This threshold of vulnerability was employed for our analysis across all three indices. The number of indices under which a census tract could be considered vulnerable was then counted; we will refer to this metric as a census tract's compound vulnerability score. Agreement of the three indices was further analyzed by grouping the census tracts into

the nine climate regions defined by California's Fourth Climate Change Assessment [42]. If a tract straddled multiple regions it was assigned to the region in which it had greater area.

## Finding communities at risk from extreme heat

In our analysis, relative and absolute extreme heat metrics were used to capture the diverse relevant forms of heat exposure. Temperature data selection and preparation matched that done in California's Fourth Climate Change Assessment [43]. We used Localized Constructed Analog downscaled Coupled Model Intercomparison Project 5 temperature data downloaded from Cal-Adapt, a government run API portal, and humidity data from the University of California San Diego's archive [44–47]. We downloaded daily maximum temperature (Tmax), minimum temperature (Tmin), and minimum relative humidity data for each tract from these sources. The four climate models highlighted in the state assessment were selected: HadGEM2-ES, CNRM-CM5, CanESM2, and MIROC5. Our study considers three time periods: historical (1950–1999), current (2006–2025), and projected (2040–2059). The current period uses data from the four climate models. All periods consider data between April 1st and October 31st, given the focus on extreme heat. We used a moderate emissions scenario only (Representative Concentration Pathway 4.5) given overlapping climate projections across scenarios over the next few decades [43]. By the end of the century, however, divergence across scenarios is more substantial. At present, global emissions and greenhouse gas mitigation efforts are not in line with the moderate trajectory, and without increasing stringency of emissions reductions, global warming and associated climate change impacts would exceed this trajectory in the longer term.

**Relative extreme heat events.** Relative extreme events were identified based on local thresholds derived from historically observed heat extremes. Historical Tmax and Tmin data provide a place-relevant snapshot of extreme heat that can be used as a reference for assessing future risk. Rather than follow the heat wave and extreme heat day definitions provided by the default on Cal-Adapt, we adopted a commonly applied approach that uses more health-relevant thresholds: a heat-wave threshold at the 95th historical percentile with a two day duration window [48–51]. A minimum and a maximum historical extreme heat threshold was calculated for each census tract by finding the 95th percentile over the historical period for daily Tmax and Tmin values.

Distinct daytime and nighttime temperature extremes in heat wave calculations were maintained to distinguish the different types of future heat wave events. There are two types of heat waves examined in this study: daytime (Tmax) and nighttime (Tmin) events. Daytime and nighttime heat waves occur when daily maximum and minimum temperatures surpass the local maximum and minimum historical threshold for two consecutive days, respectively. The total number of daytime and nighttime heat waves per year, for April 1st through October 31st over the current and projected time periods, was calculated for each climate model. Across the four climate models, the average number of heat waves per year was calculated for each heat wave type in each census tract. The change in number of heat waves, compared to the current period, was then determined.

**Absolute extreme heat events.** We also employed an absolute extreme heat definition using the heat index. Previous work has questioned the efficacy of a 105˚F heat index threshold for the state of California [16, 25]. Given that health impacts have been shown to emerge at temperatures as low as 73˚F for the coastal region of San Diego County, we used the lower bound of the heat index "extreme caution" classification, 91˚F, as a mid-range, health-relevant definition for the state [25]. In contrast to the relative heat events, a single temperature threshold was used for the entire state rather than thresholds derived from local historical data. To

find the number of absolute extreme heat events, the number of heat index days for each model, April 1st through October 31st over the projected period, was counted, averaged, and divided by the number of years in the period to get the average number of heat index events per year per tract. Heat index days were calculated using the "weathermetrics" R package [52]. Change in the number of extreme heat days between the projected period and the current period was calculated.

**Distribution of extreme heat and vulnerability.** Risk is a function of vulnerability, exposure, and hazards [10]. To develop an understanding of which communities are most at risk we overlaid extreme heat exposure and the vulnerability indices. The most at-risk tracts are those with high heat exposure and high vulnerability.

## Assessing heat-related CCI-funded interventions

We studied how California has allocated CCI-funded interventions across vulnerable groups to date. Data about CCI projects were obtained from the CCI website and included projects through November 30th, 2019 [53]. Many projects spanned multiple years and census tracts. Here, each unique combination of project identification number and tract was considered a tract intervention. Multi-year projects within a tract that had the same project identification number were considered one intervention, yielding 56,125 unique tract interventions from 2,929 unique project identification numbers. Data were further narrowed down to include heat-relevant investments, which were classified into four categories: heat, outdoor workers, greening and surfaces, and weatherization. Selection of interventions for each of these categories built from existing guidelines for systematic review searching and screening and for topic coding through qualitative content analysis [54, 55]. The search strategy included two iterations of inclusion and exclusion of projects based on keyword search terms. For the first iteration, a list of heat-specific adaptation words was created for each of the categories drawing from keyword phrases familiar to the authors and within the literature considered throughout this study. The keyword searches were done in the following CCI project data columns: "ProjectDescription," "DisadvantagedCommunityBenefitsDescription," and "OtherProjectBenefitsDescription." From this first search, we developed a table of the number of tract interventions associated with each phrase (S1 Fig). This table was reviewed and keyword phrases that returned zero results were removed from the keyword list. Phrases that returned tract interventions with subprograms that were not obviously relevant, such as the "Alternative Manure Management Program" or the "Fire Prevention Grant Program," were further scrutinized. From this analysis, a second round of inclusion and exclusion keyword phrases was developed (S1 Fig). This second iteration was used to identify CCI-funded tract adaptations for each of the heat-focused categories.

Heat-related interventions were highly variable with award amounts ranging from hundreds to millions of dollars. They spanned multiple programs such as the Low Income Weatherization, Urban Greening, and Community Air Grants programs. Because of the difficulty in selecting a single metric on which to measure intervention scope, we instead examined the number of heat-related tract interventions. First, we measured how the state of California has distributed all CCI-funded tract interventions and heat-specific tract interventions across communities based on their compound vulnerability scores. This was compared to the percent of tracts in the state that are vulnerable under any of the indices studied. Second, we analyzed how the tract interventions were allocated based on vulnerability indices. Third, we analyzed how the implementation of AB 1550 impacted delegation of heat-related CCI-funded tract interventions, seeking to understand the influence of the state's new approach for equity-oriented climate adaptation. Projects in 2015 and 2016 occurred before the passage of AB 1550

and projects in 2017, 2018, and 2019 happened after. Data on a tract's designation as low income under AB 1550 were downloaded from the California Air Resources Board website [56].

## Results

### Selection of the most vulnerable communities varies across indices

CES, HHAI, and SVI have distinct patterns of vulnerability across California's census tracts (Fig 1). CES, HHAI, and SVI are calculated based on different underlying indicators, grouped by the concept they represent in Table 1. There is agreement across the three indices that Poverty, Educational Attainment, and Linguistic Isolation are significant concepts for determining vulnerability, but the indices differ in how they measure those concepts. For example, all three indices include at least one poverty-related indicator, but HHAI and SVI consider the percentage of the population below the poverty level while CES considers the percentage of the population below two times the poverty level.

All indices are relevant to extreme heat, but their different compositions lead to contrasting assessments of vulnerability. CES is an average of two overall components: pollution burden and population characteristics. Pollution burden is captured by the concepts Air Pollution and Land Based Pollution (Table 1), which reflect twelve unique indicators for CES, such as presence of solid waste sites and facilities and pesticide use. An emphasis on pollution burden explains some of the dissimilarity between CES and the other indices. The median vulnerability index percentile values for tracts that have a pollution burden in the upper quartile are 84.3 for CES, 67.6 for HHAI, and 68.4 for SVI. Hence SVI and HHAI prioritize pollution less in classification of most vulnerable communities. Five out of sixteen of HHAI's indicators pertain to the category Environmental Conditions and only two are pollution specific. SVI has no indicators in the Environmental Conditions or Health categories.

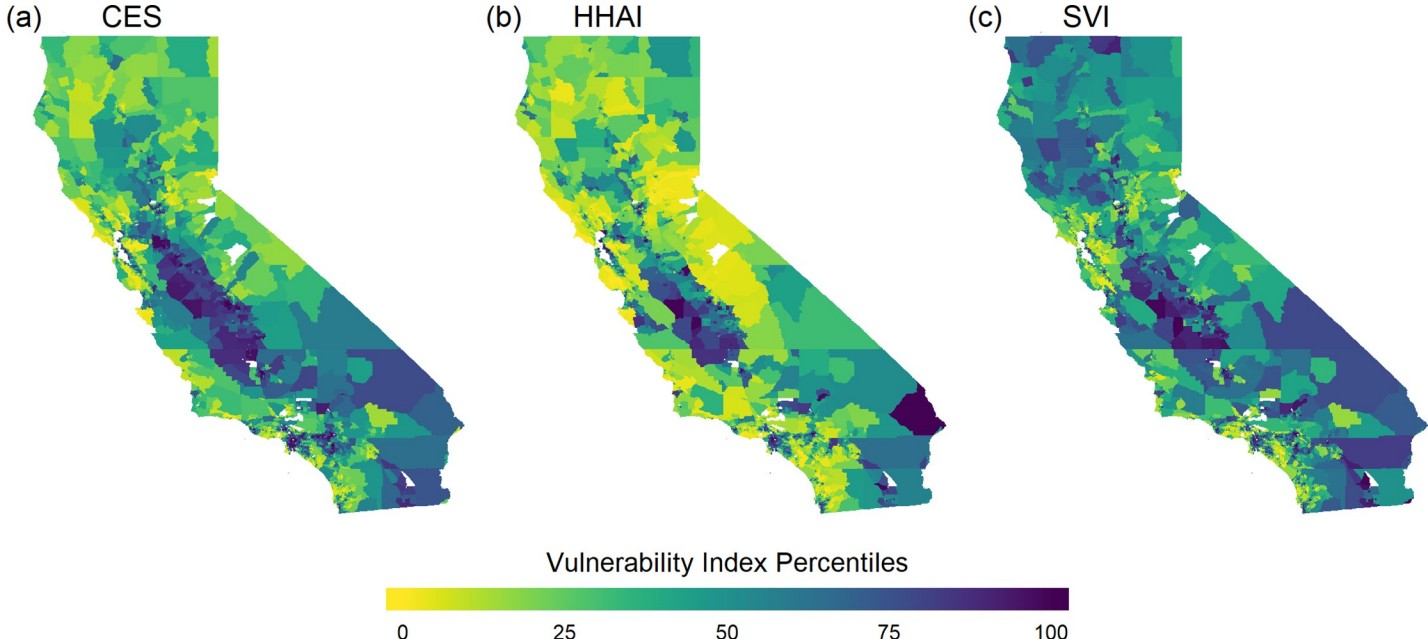

**Fig 1. Vulnerability at the census tract level differs by index.** Three vulnerability indices are plotted for each census tract in California: (a) CalEnviroScreen 3.0 (CES), (b) the Heat-Health Action Index (HHAI), and (c) the Social Vulnerability Index (SVI). In each panel, the 99[th] percentile specifies the most vulnerable census tracts under the index. Tracts that did not have reported values for any or all of the indices are white.

**Table 1. Differences across vulnerability indices.**

| Category | Concept | CES | HHAI | SVI |
|---|---|:---:|:---:|:---:|
| Economic | Poverty | X | X | X |
| | Unemployment | X | | X |
| | Outdoor Workers | | X | |
| | Housing Burden | X | | |
| | Vehicle Access | | X | X |
| | Household & Living Characteristics | | | X |
| Demographic | Educational Attainment | X | X | X |
| | Age | | X | X |
| | Civilians with a Disability | | X | X |
| | Single-Parent Households | | | X |
| | Race | | | X |
| | Linguistic Isolation | X | X | X |
| Health | Low Birth Weight Infants | X | X | |
| | Asthma | X | X | |
| | Cardiovascular Disease | X | X | |
| Environmental Conditions | Air Pollution | X | X | |
| | Surfaces & Tree Canopy | | X | |
| | Land Based Pollution | X | | |

The three vulnerability indices—CES, HHAI, and SVI—differ in their specific indicators used, which are classified here by the concepts they represent across four broad categories.

The indices differentially prioritize urban versus rural vulnerabilities. Vulnerability in communities with low population densities is ranked the highest by SVI and the lowest by HHAI (S2 Fig). However, across indices, tracts with high population densities generally are more vulnerable. Among tracts in the top decile of population density, median vulnerability index percentile values are 75.0 (CES), 86.8 (HHAI) and 78.0 (SVI), while tracts in the bottom decile of population density have median vulnerability index percentile values of 35.7 (CES), 19.5 (HHAI), and 44.0 (SVI). HHAI includes three indicators that are important to characterizing urban-specific heat vulnerabilities but are less relevant for rural communities: impervious surface coverage, tree canopy coverage, and the change in urbanized land (Table 1). SVI excludes environmental, pollution, and health related indicators, which may make it better able to characterize the drivers of vulnerability in low population density tracts (Table 1 and S2 Fig). Beyond urban versus rural heat differences, the indices do not consider regional environmental and climatic differences that influence how communities experience heat and therefore how they should adapt. For example, prior work has shown that coastal winds can play a significant role in heat exposure for the inland tracts of the South Coast [57].

Some communities are considered vulnerable regardless of the index used. 13.4% (1060) of tracts in the state are vulnerable (75th percentile or above) under all three indices despite differences in indicator selection, measurement of those indicators, and calculation methods (Fig 2). In addition, disagreement across tracts designated as "disadvantaged" likely results from regional differences in vulnerability across the state as well as differences in the overall focus of the indices. "Disadvantaged" designations for tracts vulnerable under all indices point to especially high priority communities, bolstering findings from previous work [33]. As California uses CES in designation of "disadvantaged" communities, 348 tracts that are vulnerable by both SVI and HHAI are not counted as state-designated "disadvantaged" communities. There are 429 communities that are currently eligible to receive funds for "disadvantaged"

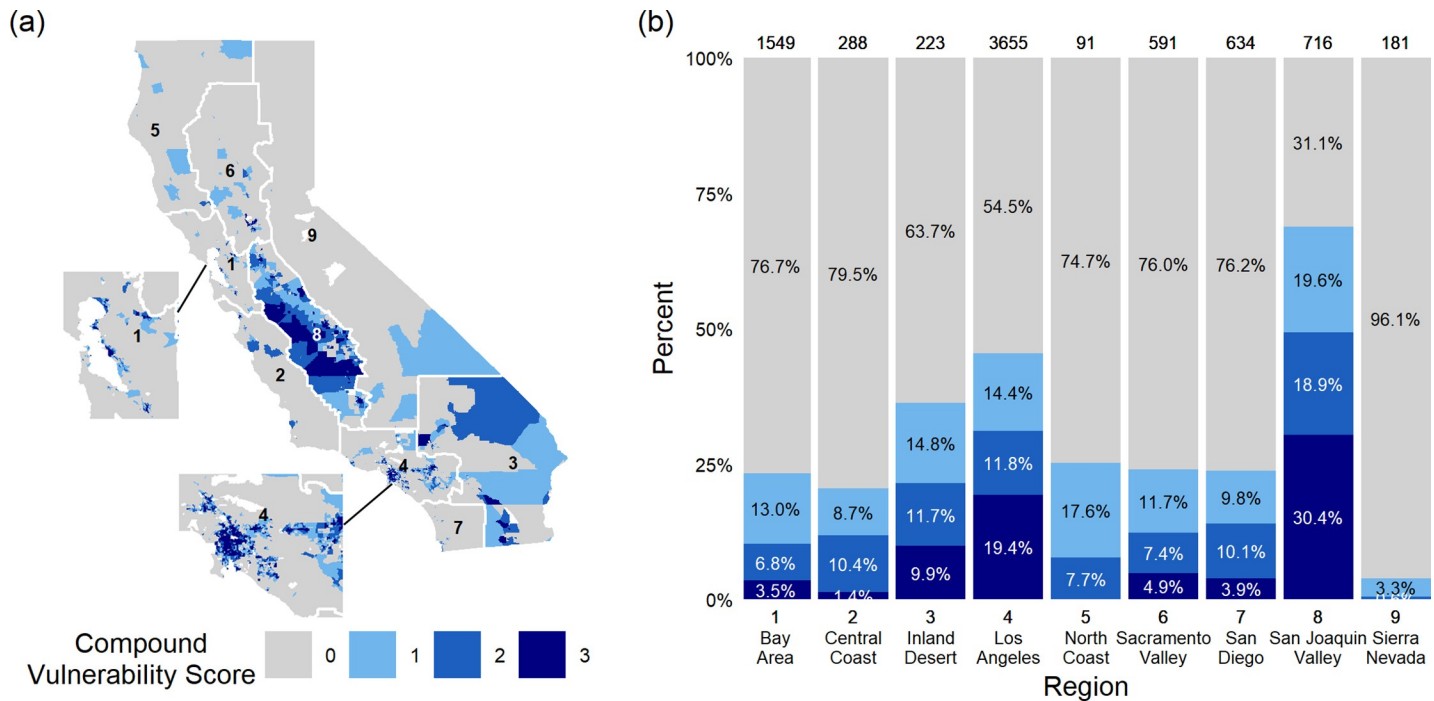

**Fig 2. Regional distribution of vulnerability across three indices.** (a) California's census tracts are shaded based on the number of indices under which they are considered vulnerable, that is, their compound vulnerability score: vulnerable under zero indices (gray), one index (light blue), two indices (sky blue) or three indices (dark blue). (b) Index agreement on the state's most vulnerable communities is depicted for each region following the color key in panel (a). Numbers above each bar represent the total number of census tracts for the region.

communities that are vulnerable under CES alone. This number represents 21.6% of the tracts currently considered "disadvantaged" by the state of California.

The most vulnerable tracts in the state do not occur equally across regions. Some regions hold a disproportionate share of the state's most vulnerable communities (Fig 2). 49.3% of tracts in the San Joaquin Valley, 31.1% of tracts in the Los Angeles and 21.5% of tracts in the Inland Desert regions are considered vulnerable by two or more indices. The Los Angeles region also accounts for the highest number of tracts vulnerable under all indices: 708. Conversely, only 7.7% and 0.6% of tracts in the North Coast and Sierra Nevada regions are considered vulnerable by two indices, respectively. No tracts in these regions are considered vulnerable by all three, and of the tracts that are designated as vulnerable, SVI is the prevailing index for the designation (S3 Fig). Census tracts that are the least vulnerable (bottom quartile of vulnerability indices) are concentrated in the Los Angeles (33.8%), the Bay Area (32.4%), and the San Diego (13.8%) regions.

## Variations in exposure and vulnerability metrics shape extreme heat risk calculations

Most of the state will experience increases in extreme heat, but the form of future extreme heat exposure will vary across communities (S4 Fig). The median number of daytime and nighttime heat waves projected to occur across census tracts from 2040–2059 is 10.3 and 17.9 events per year, respectively. This translates to 4.3 more daytime events and 9.1 more nighttime events per year than the current period, on average across census tracts (S4 Fig). Daytime events will largely impact the inland regions while nighttime exposure will increase along the coasts. Previous work has shown that, for the heat wave probability density functions, geographic

variations in tail length and variance lead to such differing daytime and nighttime heat wave predictions [51]. Average heat index days per year will increase to a median value of 44.3 events across census tracts, an average bump of 11.6 more days per year than the current period (S4 Fig).

The relationship between future heat exposure and vulnerability varies based on the exposure metric and vulnerability index employed. Compared to the rest of the state, California's most vulnerable communities are projected to experience a larger increase in daytime heat waves and heat index days (Fig 3A and 3C). This relationship is most pronounced for vulnerability measured with SVI or CES. Vulnerable communities are also projected to experience more nighttime heat waves, but the increase is more pronounced in less vulnerable communities (Fig 3B). Communities of all levels of vulnerability defined by HHAI are projected to experience similar changes in heat exposure moving forward. One reason for this divergence is that HHAI considers heat adaptive capacity indicators that are not present in the other indices, such as the presence of outdoor workers or tree canopy coverage. These results present general trends for the entire state. However, there remain multiple highly vulnerable communities that face elevated exposure to nighttime events. The choice of index influences how risk is measured across census tracts in California.

## Equity-oriented CCI-funded interventions in California

To date California has directed most of the heat-related CCI-funded investments to its most vulnerable communities (Fig 4A). This analysis considers 56,125 CCI-funded tract interventions. Under our search criteria, we identify 2,503 of those as unique heat-related adaptations across four project categories: heat, greening and surfaces, outdoor workers, and weatherization. 323 tract interventions are duplicates across these categories. Across all heat-related project categories considered, 73.4% of tract interventions went to communities with a compound vulnerability score of two or three. In comparison, as indicated by the red line, 37.6% of tracts in the state are considered vulnerable under at least one index. 4.5%, 7.8%, and 7.3% of tract interventions went to communities not considered vulnerable under any index for the heat, greening and surfaces, and weatherization categories, respectively. For all CCI-funded tract interventions (far-right column of Fig 4A), the allocation of interventions across tracts with varying compound vulnerability scores closely matches the distribution of tracts with those scores across the state.

Unsurprisingly, designation as "disadvantaged" under CES is a major factor guiding which vulnerable communities received CCI-funded tract interventions (Fig 4B). Communities in the upper quartile of vulnerability under CES received 82.4% of heat-related CCI-funded tract interventions. Communities similarly vulnerable under HHAI, the heat specific index produced by the state, received only 65.3% of heat-related tract interventions. Of the interventions in HHAI-vulnerable tracts, the vast majority went to communities designated as vulnerable under both indices; only 8.7% of HHAI interventions went to communities that were vulnerable under HHAI but not CES. Conversely, 27.7% of the 2,063 heat-related adaptations in CES-vulnerable communities went to tracts vulnerable under CES but not HHAI. 1,279 interventions went to tracts vulnerable under all three of the indices. Although the discrepancy between HHAI and CES persisted across all CCI-funded interventions, it is less pronounced. 25.3% and 30.8% of tract interventions went to communities vulnerable under HHAI and CES, respectively. Tracts that are vulnerable under CES but not HHAI received 4.0 times the number of heat-related interventions as tracts vulnerable under HHAI but not CES. This number shrinks to 1.8 when considering all CCI interventions implemented by the state.

AB 1550 minimally reallocated interventions to communities with diverse forms of vulnerability (Fig 5). AB 1550 was created to redistribute adaptations to low income communities

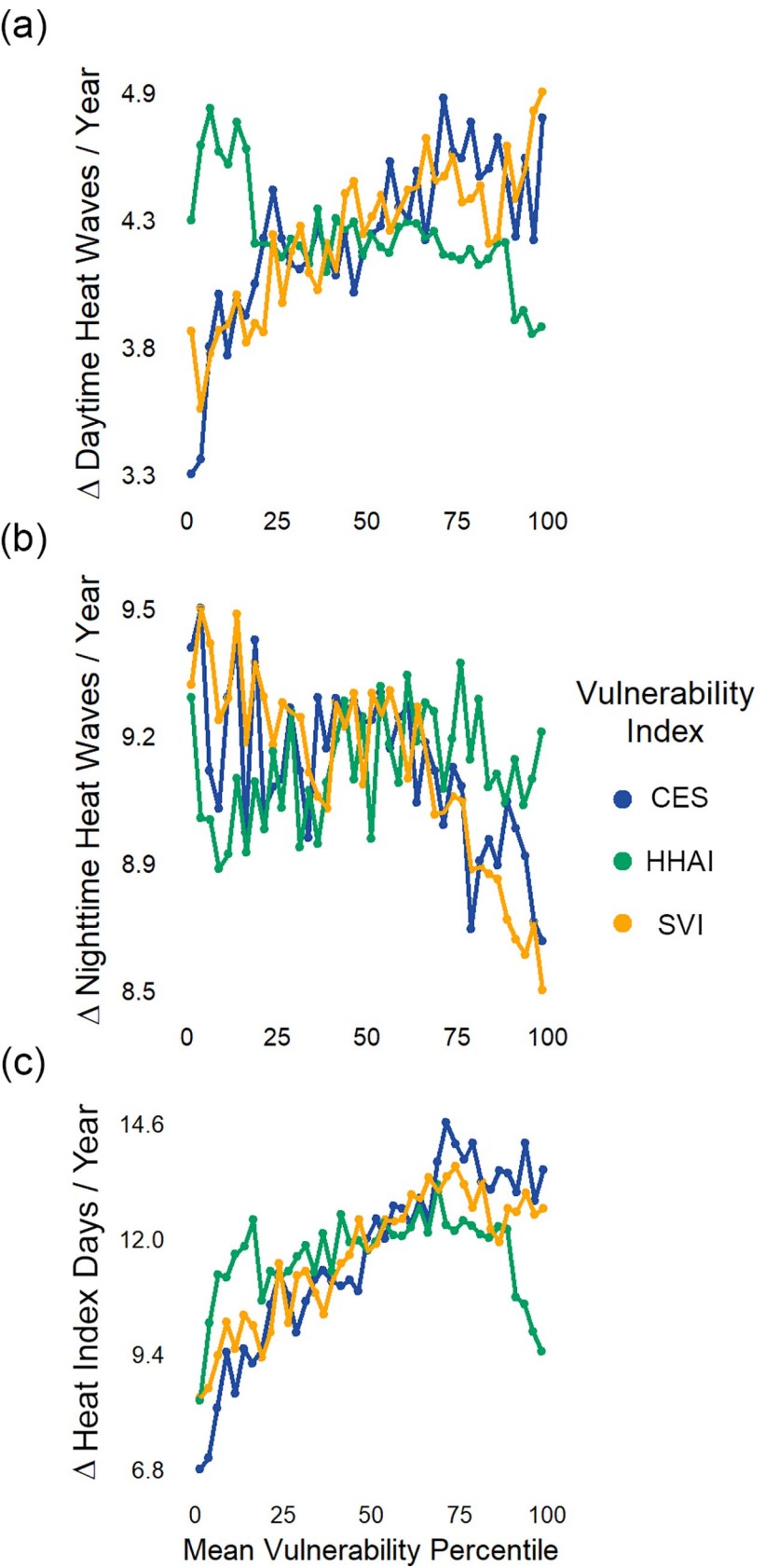

**Fig 3. Trends in extreme heat exposure vary by vulnerability index and exposure metric.** Data points represent change in heat exposure for census tracts binned by vulnerability index percentile (approximately 200 tracts per bin of 2.5 percentile width). Each color corresponds to a vulnerability index: CES (blue), HHAI (green), and SVI (yellow). Exposure is measured as the increase in extreme heat events from the current period to the projected period for three metrics: daytime heat waves (a), nighttime heat waves (b), and heat index days (c).

that do not qualify as "disadvantaged" under CES. After AB 1550, census tracts considered vulnerable under CES received a 4.9 percentage point decrease in heat-related CCI-funded tract interventions (Fig 5A) and a 5.1 percentage point increase in all CCI-funded tract interventions (Fig 5B). Low income communities that are not disadvantaged under CES saw a 2.2 percentage point increase and a 2.7 percentage point increase in intervention fractions, respectively. Tracts that are neither "disadvantaged" nor low income saw a 2.7 percentage point increase for heat-related interventions and a 7.8 percentage point decrease for all tract interventions after AB 1550. Hence, intervention allocation experienced minor changes after the enactment of AB 1550.

## Discussion

Vulnerability indices are commonly used to prioritize adaptation needs and investments. However, the choice of an index invariably involves trade-offs and challenges. Here, we use extreme heat adaptation in California as a case study to analyze implications for implementing

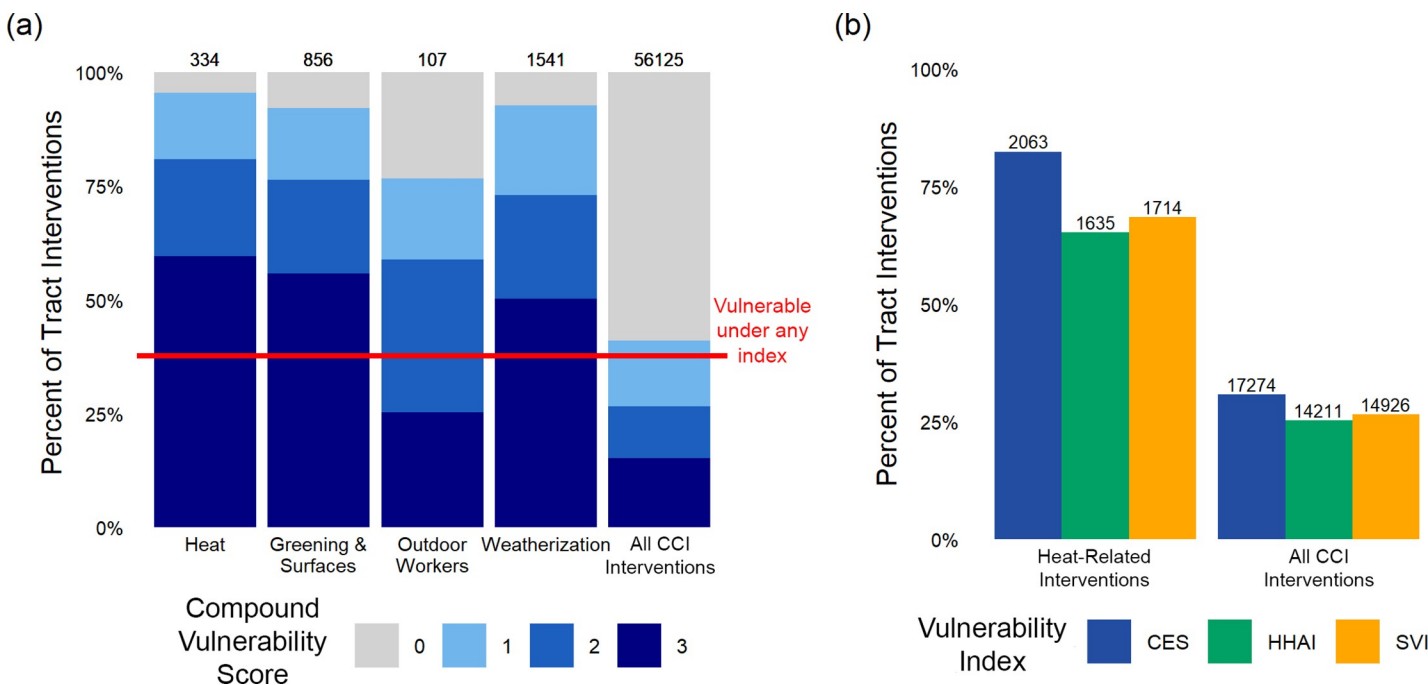

**Fig 4. CCI-funded tract interventions across census tracts differ by project focus and vulnerability.** Plots explore the relationship between vulnerability and CCI project allocation across heat-related interventions and all interventions done to date. Panel (a) displays the percent of tract interventions across tracts with different compound vulnerability scores. The compound vulnerability score in (a) is equivalent to that used in Fig 2; the score corresponds to a tract being considered vulnerable under zero (grey), one (light blue), two (sky blue) or three (dark blue) vulnerability indices. The first four bars ("Heat," "Greening & Surfaces," "Outdoor Workers," and "Weatherization") are considered the heat-related tract interventions and have some overlap among them. The last bar ("All CCI Interventions") considers all interventions done by the state, including the interventions encompassed in the previous four bars. The red line represents the percent of census tracts in the state that have a compound vulnerability score of one or greater. Panel (b) displays the percent of tract interventions for heat-related interventions and all CCI interventions in vulnerable communities, as defined by each index. Vulnerability index colors match that of Fig 3. Numbers above the bars in both panels represent the total number of interventions in tracts designated as vulnerable by that index.

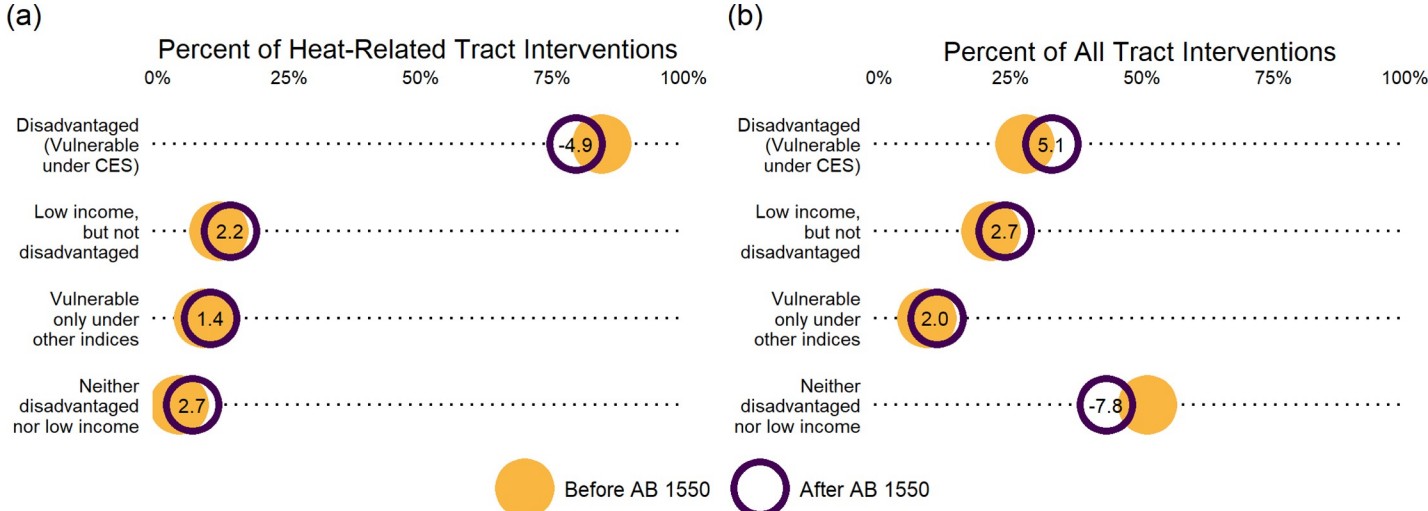

**Fig 5. Change in the allocation of CCI-funded projects after the passage of AB 1550.** Plots display the percentage of CCI-funded tract interventions for different subsets of the population before (yellow) and after (purple) the implementation of AB 1550. Panel (a) considers heat-related tract interventions and panel (b) considers all tract interventions. Both panels consider the same population groupings. Tracts that are listed as both disadvantaged and low income by the state are included only as "Disadvantaged (Vulnerable under CES)". Communities that are "Neither disadvantaged nor low income" are not considered vulnerable under CES nor are they low income; they could, however, be considered vulnerable by HHAI and/or SVI. Numbers in the circles represent the percentage change of heat-related projects after the passage of AB 1550.

equity-oriented climate adaptations among most vulnerable communities. This study finds that by using CalEnviroScreen 3.0 to designate a census tract's status as "disadvantaged," the state does not fully capture community vulnerability to extreme heat, as might be expected. Nuances in the form of extreme heat further obfuscate where adaptation need is greatest given diverging patterns between the type of exposure and vulnerability. We show that, overall, the state has successfully targeted CCI-funded tract interventions to its most "disadvantaged" communities, but that climate change will likely increase the importance of a holistic approach for adaptation investments that reflects the diversity of the regions, the urban-rural gradient, and the scales of implementation in the state. This work demonstrates that targeted indices (HHAI or CES) and a broader index (SVI) differentially weigh and represent diversity across the large state of California. Therefore on-the-ground validation and integrative applications are vitally important for such quantitative measurements of vulnerability.

## Using vulnerability indices to guide equity-oriented adaptation

The state is currently using CES to designate communities as "disadvantaged" and therefore included in the GGRF's disadvantaged community budget. The results of this study highlight the strengths and limitations of this approach to guiding equity-oriented adaptation. First, our analysis demonstrates that one strength of using vulnerability indices is in highlighting which communities might benefit from additional investment (Fig 4B). Vulnerability indices in adaptation work have been shown to provide transparent, comparative analysis in resiliency building work [58]. This study also demonstrates their value as a starting point for further investigation, a finding consistent with previous research [59]. Second, the single index that is heat specific, HHAI, selects 653 different communities compared to the current index used by the state (Figs 1 and 2). Third, the state has successfully allocated the majority (51.1%) of its heat-related adaptations to communities that are vulnerable under all three indices (Fig 4A). The state passed AB 1550 as a response to concerns about using CES alone to allocate GGRF

monies. However, our results demonstrate that this policy has only modestly changed the allocation of CCI-funded projects among California's communities (Fig 5). The state has successfully directed the majority of CCI-funded heat-related adaptations to its most vulnerable communities, but current methods for broadening vulnerability designations among low income communities have had little effect.

There are limits to the effectiveness of using a single index to guide a fund with projects and communities as diverse as those under the CCI program. At present, CES is and has been used across state agencies for a variety of purposes beyond the GGRF, including but not limited to an abandoned underground storage tank initiative by the State Water Resources Control Board, allocation of renewable energy by the California Public Utilities Commission, and a project for active transportation by the California Department of Transportation [11]. Prior research has challenged the use of indices for broad and multifaceted applications [59] that fail to integrate contextual differences [60] or to have a clearly defined purpose [61]. Further, race-based indicators, which have been shown to be uniquely associated with environmental justice due to structural discriminatory legacies, were removed to make the tool legally viable for multiple agencies [62]. Although CES is responsive to environmental injustices, by itself it does not fully capture the climate justice adaptation problems that come with escalating extreme heat exposure. This work demonstrates that some indices may characterize some regions better than others, and that there is likely no "correct" index that can balance the unique geospatial drivers of vulnerability. A general, statewide tool could be useful for high-level screening, but context-specific tools that can better capture and reflect the nuances of the specific problem in question and local variabilities are also needed.

Overall, we find, unsurprisingly, that the "disadvantaged" designation is sensitive to index choice. Relying on a single index to guide equity-oriented adaptation may be too restrictive. Complementing existing approaches is important to fully assess community vulnerability to extreme heat. Here we have considered ways to increase robustness in vulnerability screening through intersection of CES with other relevant indices. Previous work on social vulnerability indices has aimed to address measurement uncertainty and bias by varying aspects of index construction via Monte Carlo simulations to generate a "frequency distribution of social vulnerability ranks for each tract" [28]. One study showed that a multiplex network analysis could be used to strengthen indicator selection by considering their relative weights, interdependencies, and interactions [63]. Another assessed and validated index design choices through a global sensitivity analysis [64]. Beyond quantitative checks, qualitative methods build robustness into community screening when conducting equity-oriented adaptation work. "Ground-truthing" and implementing community-based participatory research methods are essential complements and provide validity checks to data-based, quantitative representations of vulnerability [33–35]. Other techniques such as case studies, expert elicitation, etc. can provide context-specific information and illuminate dynamics and hierarchical structures that might be over looked under quantitative approaches alone [58]. CEJA's report provides further recommendations for how to supplement and expand CalEnviroScreen specifically for different policy endeavors [11].

## Implications for other environmental hazards

Although this study focused on extreme heat, there are takeaways relevant to other hazards. First, across hazards, relying on a single quantitative vulnerability index without complementary analyses to guide adaptation decisions is likely insufficient. We demonstrated this finding for heat, but previous work has come to similar conclusions for sea level rise [65], flooding [60], and wildfires [66]. A balance of data sources and methods for assessing vulnerability is

needed to capture the richness and nuances across communities. Second, exploring the extent to which equity-oriented adaptations are needed for other hazards could be valuable. In this study, we found that increases in extreme heat under climate change will significantly impact vulnerable communities in California (Fig 3). Thus, there are clear opportunities with extreme heat for implementing solutions for those most in need and for thereby addressing environmental injustices. Recent work has shown that in inland communities, but not coastal communities, "economically disadvantaged" groups are more often found in flood zones than their wealthier counterparts [67]. Future research could explore whether such a relationship exists for other hazards such as wildfire. Third, addressing pollution justice concerns may not be equivalent to equitable adaptation to climate hazards. In this work we found that using CES, an index that was developed to consider cumulative pollution burden in addition to more traditional indicators of social vulnerability, did not result in equal prioritization of communities that will likely be at the front line of heat-related impacts in a changing climate (Figs 2 and 4). Nevertheless, the drivers of inequities in pollution exposure are likely linked to inequities in resource access. Previous work demonstrated that historic zoning practices such as redlining have been associated with increased urban heat island effect and pollution [68–70]. Understanding this link between pollution burden, elevated exposure, and equitable adaptation is critical across hazards.

## Limitations and future research

Vulnerability is difficult to capture fully given its complexity and nuances. As a tension in this work, it is challenging to apply concepts of vulnerability consistent with existing literature and policies while remaining cognizant of and prioritizing the ongoing feedback from communities. Here we relied on language and scientific precedent set by California and the IPCC. However, California environmental justice communities often feel disempowered by these traditional framings of vulnerability [11, 13]. Other limitations of this work include the use of only two other vulnerability indices and examination at the census tract level, which may mask important household-level differences. Further, HHAI is not a percentile-based index so our framing of the "disadvantaged" threshold was for comparison purposes only. Finally, the diversity of impact and award amount of CCI-funded projects was not considered in the analysis.

Future research is crucial to understand best practices for implementing equity-oriented adaptation. More work could be done in understanding how to prioritize and implement adaptation projects in a state with hazards and communities as diverse as California's. Scholarship employing community-engaged methods [71] is needed to research framings and language that reflect how communities want to be represented in policy and research. Limitations and opportunities in scales of implementation are also unclear. Research is needed on how to balance the intricacies of risk with techniques that allow stakeholders to monitor and quantify the impacts of equity-oriented adaptation projects. Future work could also consider the distinctions between rural and urban vulnerability and their unique adaptation needs. Further, additional research could study conflicting forms of vulnerability and adaptive capacity across hazards, such as tree coverage reducing extreme heat risk while potentially increasing wildfire risk. Finally, future work could establish best practices for on-the-ground index validation in support of fine-scale and large-scale implementation.

## Conclusion

Many governments and stakeholders are interested in equitable climate adaptation, but widescale programs in the implementation phase are just beginning to emerge. Using extreme heat

adaptations in California as a case study, this work analyzed the role and limitations of quantitative measurements of vulnerability guiding equity-oriented adaptation policies in current practice. First, we found that using a single index to select "disadvantaged" communities, a common practice, is restrictive and not representative of the nuances of environmental and climate justice. California has successfully targeted most of the projects from the California Climate Investments program to its most vulnerable communities, but several communities that would likely benefit from heat-related adaptations have been overlooked. Fortifying screening tools to take nuances into account could help strengthen the program moving forward.

Extreme heat has disproportionately impacted certain communities, and these impacts are expected to continue to be unequally distributed under a changing climate, perhaps even more so. Understanding of how to equitably adapt to extreme heat is emerging and advancing. Although this study is region specific and heat specific, lessons from this work about how we consider vulnerability can be applied across different locations, scales, and hazards. There are many opportunities to increase resiliency and address inequities across communities.

## Supporting information

**S1 Fig. Keyword table for selection of heat-related CCI interventions.** The table displays the first and second rounds of the keyword phrases used to screen for heat-related CCI tract interventions. Both rounds are split up into four adaptation relevant categories: heat specific, outdoor workers, greening and surfaces, and weatherization. The first round (left box) is the first iteration of phrases considered. The total number of interventions found by each phrase is listed to its right. The second round includes the inclusion and exclusion keyword phrases that were determined to produce heat relevant interventions (right box).
(TIF)

**S2 Fig. Vulnerability percentiles are greater in tracts with larger population densities.** (a) Average vulnerability percentile (CES, HHAI, SVI) is plotted by log average population density of tracts (people per square kilometer, bin width of 2.5 percentiles, log scale). (b) This plot shows the frequency of different population densities across California's census tracts (people per square kilometer, log scaled x-axis).
(TIF)

**S3 Fig. Indices vary in how they weigh vulnerability across California's regions.** Each boxplot represents the distribution of census tract vulnerability percentiles for a region. Here we consider three indices: CES (blue), HHAI (green), and SVI (yellow). The red line represents the median.
(TIF)

**S4 Fig. Projected changes in extreme heat in California.** Plots display the increase in daytime heat waves (a), nighttime heat waves (b), and heat index days (c) for the state of California between the current (2006–2025) period and the future (2040–2059) period. For each metric of extreme heat, the projected change under RCP 4.5 is indicated based on the average of four climate models: HadGEM2-ES, CNRM-CM5, CanESM2, and MIROC5.
(TIF)

## Author Contributions

**Conceptualization:** Lynée L. Turek-Hankins, Miyuki Hino, Katharine J. Mach.

**Formal analysis:** Lynée L. Turek-Hankins, Miyuki Hino, Katharine J. Mach.

**Investigation:** Lynée L. Turek-Hankins.

**Methodology:** Lynée L. Turek-Hankins, Miyuki Hino, Katharine J. Mach.

**Supervision:** Miyuki Hino, Katharine J. Mach.

**Validation:** Miyuki Hino, Katharine J. Mach.

**Visualization:** Lynée L. Turek-Hankins.

**Writing – original draft:** Lynée L. Turek-Hankins.

**Writing – review & editing:** Lynée L. Turek-Hankins, Miyuki Hino, Katharine J. Mach.

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
