## [Decision Letter · Decision Letter 0]

17 Jul 2020

PONE-D-20-13248

Equity-Oriented Adaptation to Extreme Heat in California

PLOS ONE

Dear Dr. Turek-Hankins,

Thank you for submitting your manuscript to PLOS ONE. After careful consideration, we feel that it has merit but does not fully meet PLOS ONE’s publication criteria as it currently stands. Therefore, we invite you to submit a revised version of the manuscript that addresses the points raised during the review process. The three reviewers were in agreement that the manuscript needed minor revisions before it could be accepted for publication. Please address their specific comments, which appear to be constructive and should improve the final version of the paper, in your resubmission.

We look forward to receiving your revised manuscript.

Kind regards,

Joe McFadden

Academic Editor

PLOS ONE

Journal Requirements:

2.  We note that Figures 1, 2 and supporting figure 1 in your submission contain map images which may be copyrighted. All PLOS content is published under the Creative Commons Attribution License (CC BY 4.0), which means that the manuscript, images, and Supporting Information files will be freely available online, and any third party is permitted to access, download, copy, distribute, and use these materials in any way, even commercially, with proper attribution. For these reasons, we cannot publish previously copyrighted maps or satellite images created using proprietary data, such as Google software (Google Maps, Street View, and Earth). For more information, see our copyright guidelines: http://journals.plos.org/plosone/s/licenses-and-copyright.

a) You may seek permission from the original copyright holder of Figure(s) [#] to publish the content specifically under the CC BY 4.0 license. 

Reviewers' comments:

Reviewer's Responses to Questions

**Comments to the Author**

1. Is the manuscript technically sound, and do the data support the conclusions?

Reviewer #1: Yes

Reviewer #2: Yes

Reviewer #3: Yes

2. Has the statistical analysis been performed appropriately and rigorously? 

Reviewer #1: Yes

Reviewer #2: Yes

Reviewer #3: Yes

3. Have the authors made all data underlying the findings in their manuscript fully available?

Reviewer #1: Yes

Reviewer #2: Yes

Reviewer #3: Yes

4. Is the manuscript presented in an intelligible fashion and written in standard English?

Reviewer #1: Yes

Reviewer #2: Yes

Reviewer #3: Yes

5. Review Comments to the Author

Reviewer #1: Review of: “Equity-Oriented Adaptation to Extreme Heat in California”

This paper uses three vulnerability indices for California to show how census-tract assessments of vulnerability are sensitive to the choice of vulnerability index. The paper further investigates how adaptation projects are distributed across census tracts, and shows how well the projects are meeting the target of helping vulnerable populations. The findings reveal that relying on a single index (CES) in the distribution of these projects could leave out some vulnerable communities that are shown to be vulnerable by other indices other than the CES. Overall, I found this paper to be extremely well written and informative. This is an important and timely topic and should be of interest to many. This work clearly has applications to climate change adaptation and planning in California. I see no major issues with the article, and I think it is acceptable for publication after addressing a few minor comments listed below.

Minor Comments:

Line 190: RCP 4.5 is a low-emissions scenario that assumes aggressive action to curb CO2 emissions. Under this scenario, radiative forcing from CO2 would stabilize before the end of the century, however, we are not on track to meet this goal. It would be useful to mention that the projections used in this study are likely to underestimate future heat impacts due to climate change, especially at the end of the century.

Line 195: What is the source of the Tmin and Tmax temperature data?

Lines 460-463: You mention that the HHAI is the only index that is heat specific. This is an important point, but it is unclear from Table 1 how heat is included in this index.

Reviewer #2: Review of Equity-Oriented Adaptation to Extreme Heat in California

Overview. This is a methodologically-oriented article that develops and applies a rigorous, if not entirely novel, comparative approach to three different heat vulnerability indices in California. It makes the important point that models matter and that designation of areas as vulnerable to extreme heat depends greatly on the type of index used. The article applies this analysis to a critique of state climate adaptation funding.

Strengths:

The methodology has a number of strengths. It selects three relevant indices that can be used for assessing heat vulnerability: both because of their specific sets of indicators and spatial analysis approach and because of their alignment with public policy initiatives. It provides a very clear and systematic review of each indicator and an elegant way of comparing across all three. The analysis of which census tracts are/are not indicated as vulnerable in each of the three indices comes across in a compelling way. Most of the maps and graphs are clear and help make the paper’s argument in a compelling way. Their analysis of the differences in identifying vulnerable to heat risks can have significant policy impacts and implications for distribution of public funding. The article is very timely as billions of dollars of public investments in California to address health issues related to climate change and other environmental hazard are being made based on the designation of vulnerable communities. On the note of policy, figure 5 provides a very useful critique of AB 1550 by showing the minor degree of change in climate investments since its passage.

Areas for Improvement:

The paper provides an excellent review of the literature on heat vulnerability but does not situate itself in the literature of the construction and comparison of indicators (with the exception of references 54 and 55—but these are referenced only in passing and only in the conclusion. This discussion should be brought up into the lit review and further developed. This detracts from the value of the paper in contributing to this growing field. Adding this component to the literature review could help the authors identify what is novel about their methods and how they could be fine-tuned to push the boundaries of the field .Two pieces that may be useful in this regard are:

• James Sadd et al. 2014. R9 RARE Final Project Report Title: Partnering with Environmental Agencies and Communities to Pilot Use of the Environmental Justice Screening Method (EJSM) Cumulative Impacts Tool. US EPA Region 9.

• Liévanos, Raoul. 2018. Retooling CalEnviroScreen: Cumulative pollution burden and race-based environmental health vulnerabilities in California. International journal of environmental research and public health, 15(4), 762.

One critique of CES (that can apply to the SVI and HHAI) is that as a state-wide tool, it is less sensitive to regional variations. That is, heat vulnerability can be the result of different factors in different environments and also require different adaptation strategies. The paper takes this up at line 322, but this section would be expanded upon. What is it about each tool that produces this pattern? Breaking out their analysis by region (exploring whether the tools different in their ability to identify high heat vulnerable communities) would provide some useful data for state agencies charged with using them.

The article notes that CES is not a heat vulnerability/ climate change risk tool but could make a stronger point about the challenges for the state in using a general purpose tool for a specific purpose. This is increasingly challenging as CES has been tasked to cover more and more policy areas, from toxic waste management, to air quality management, to drinking water. The article could help argue that while holistic tools like CES are very valuable for directing resources to address overall vulnerability/ disadvantage, more context-specific tools (or perhaps customized modules) will be needed.

The discussion section acknowledges that no matter how sophisticated, tools like CES, SVI, and HHAI are, they are best used as screening methods (indeed, there is a tool by developed for the California Air Resources Board called the Environmental Justice Screening Method – see above report by Jim Sadd) that can flag areas that require more in-depth and ground-level assessment. While this is expensive to implement, it is important for public agencies not to rely solely on these “30,000 foot” models and instead commitment to on-site ground-truthing. This point could made more strongly and in the introduction instead of only at the end of the paper.

Finally, the paper does not directly address of equity-oriented adaptation” as promised in its title. That is, it does not study adaptation policies or practices per se, but only the tools that can help inform adaptation strategies. Because this would require a significant design of the paper, I would suggest changing the title as opposed to doing too much to align with it. However, the paper could discuss the ways in which these tools serve to direct funds to certain kinds of adaptation projects.

Specific notes;

Lines 79-80. The paper makes the important point that many community advocates dislike the stigmatizing quality of terms such as disadvantaged or vulnerable. Could the authors suggest some alternatives?

Line 190: Define “Representative concentration pathway 4.5”

Line 220: Why was “a single temperature threshold was used for the entire state rather than thresholds derived from local historical data?” What would the analysis look like if it also used local data? This could get at the regional variation issue raised above.

Line 277: Why are “vulnerability in communities with low population densities is ranked the highest by SVI and lowest by HHAI?” This would be a great opportunity dig into the reasons behind the differences as opposed to primarily reporting on them as the article does now.

Line 314. Can the authors explain why only “13.7% (1085) of tracts in the state are vulnerable (75th percentile or above) under all three indices?” This seems to be one of most important points of the paper can use more explanation.

Line 449: The authors call for “a holistic approach for adaptation investments.” I agree this is an important point—but what would this look like? How can they use their analysis to make this point more rigorously?

Reviewer #3: The paper uses vulnerabilty estimates from three different indices as well as large scale climate data to assess spatial patterns of variability and forecasts for extreme heat frequency and severity. In general, the paper is well written and I enjoyed reading it. I have two general comments below and some textual comments below that.

The paper uses quite a bit of jargon and doesn't define a lot of important terms in terms of their analysis (e.g. vulnerability, index, community). Index is used to refer to both heat indices (which combine meteorological variables) and PCA-based weighting schemes to make an abstract assessment of risk. They include glossary definition in text, but need to better define important terms in a way that is relevant to the paper.

In general, the abstract is a bit difficult to read and could use more direct statements about what the paper is going to do. Does CalEnviroScreen 3.0 not consider variables that correlate with heat? What are the two alternative indices? The line "Only 13.7% of communities are vulnerable across all three vulnerability indices studied." seems to indicate that the indices are mutually exclusive?

line 68: What does "community screening" mean here? Does that mean risk assessment at the community level? Is that sociodemographic or climate-focused? This may benefit from a bit more explanation.

line 90: May want to also include that heat death is chronically underreported. Gubernot et al 2013 Int J Biometerol & Riley et al 2018 Int J Environ Res Public Health

line 138: Can you clarify what "intersected" means for this study? Does this mean that you applied these indices with different heat thresholds? Need more clarity here.

line 218: 91F is a very low threshold - does this come from one of the cited studies? 91 may be relevant to particularly vulnerable sub-groups (e.g. the elderly and the immunocompromised), but I think you need to better justify why you chose such a low threshold, particularly when much of California is relatively dry.

line 279: It seems like tracts with high population densities are less vulnerable on average compared to low density areas? Especially if we don't include Northern California (which is subject to less extreme heat in the first place). The next few lines clarify this a bit, but I think it still could be more clear.

Table 1: This table should be moved earlier in the paper - this would help to clarify the heat-index vs. vulnerability index issue mentioned earlier.

line 345-347: It would be helpful to posit a reason for day/night and inland/coastal differences in heat wave frequency. This plays into your findings for increased heat wave frequency in vulnerable communities as income generally increases towards the coast in California.

line 449: The use of "holistic" here is vague - this would benefit from more explanation.

line 476: Are all of these indices linear? You've normalized them and then selected "disadvantaged" communities based on a percentile threshold. That may not be valid if they come with specific numeric threshold for identifying disadvantaged communities. You may want to add this to the limitations section.

6. PLOS authors have the option to publish the peer review history of their article (what does this mean?). If published, this will include your full peer review and any attached files.

Reviewer #1: No

Reviewer #2: **Yes: **Jonathan K. London

Reviewer #3: No

---

## [Author Response · Author response to Decision Letter 0]

29 Aug 2020

Please see the attached document "Response to Reviewers" for a detailed discussion of updates and responses to the reviewer comments on the manuscript.

---

## [Editor Report · Decision Letter 1]

5 Oct 2020

Risk screening methods for extreme heat: implications for equity-oriented adaptation

PONE-D-20-13248R1

Dear Dr. Turek-Hankins,

We’re pleased to inform you that your manuscript has been judged scientifically suitable for publication and will be formally accepted for publication once it meets all outstanding technical requirements.

Kind regards,

Joe McFadden

Academic Editor

PLOS ONE

---

## [Editor Report · Acceptance letter]

12 Oct 2020

PONE-D-20-13248R1 

Risk screening methods for extreme heat: implications for equity-oriented adaptation 

Dear Dr. Turek-Hankins:

I'm pleased to inform you that your manuscript has been deemed suitable for publication in PLOS ONE. Congratulations! Your manuscript is now with our production department. 

Kind regards, 

on behalf of

Dr. Joe McFadden 

Academic Editor

PLOS ONE